# Electromagnetic polarization-controlled perfect switching effect with high-refractive-index dimers and the beam-splitter configuration

Ángela I. Barreda[1], Hassan Saleh[2,3], Amelie Litman[2], Francisco González[1], Jean-Michel Geffrin[2] & Fernando Moreno[1]

Sub-wavelength particles made from high-index dielectrics, either individual or as ensembles, are ideal candidates for multifunctional elements in optical devices. Their directionality effects are traditionally analysed through forward and backward measurements, even if these directions are not convenient for in-plane scattering practical purposes. Here we present unambiguous experimental evidence in the microwave range that for a dimer of HRI spherical particles, a perfect switching effect is observed out of those directions as a consequence of the mutual particle electric/magnetic interaction. The binary state depends on the excitation polarization. Its analysis is performed through the linear polarization degree of scattered radiation at a detection direction perpendicular to the incident direction: the beam-splitter configuration. The scaling property of Maxwell's equations allows the generalization of our results to other frequency ranges and dimension scales, for instance, the visible and the nanometric scale.

[1] Group of Optics, Department of Applied Physics, University of Cantabria, Cantabria 39005, Spain. [2] Aix-Marseille Univ, CNRS, Centrale Marseille, Institut Fresnel, Marseille, France. [3] Centre Commun de Ressources en Microondes CCRM, 5 rue Enrico Fermi, Marseille 13453, France. Correspondence and requests for materials should be addressed to F.M. (email: morenof@unican.es).

One of the most active research branches regarding the interaction of electromagnetic radiation with matter is that of its scattering by particles smaller than the wavelength of the incident radiation. Over many years, it has undergone vigorous investigation leading to applications in areas as diverse as health, material analysis, communications, and so on[1]. For the case of metallic materials, when an impinging radiation illuminates a small metallic particle, the electronic plasma oscillates with the same frequency as the incident radiation. When resonant conditions are achieved, localized surface plasmons are generated. These coherent oscillations of the metallic free electrons depend on the optical properties of the particle, its surrounding medium and also on the particle size and shape and the wavelength of the incident radiation[2]. When localized surface plasmons are generated, the energy of the impinging radiation is transferred to free electrons, which oscillate at maximum amplitude, and enhancements of the electric field are observed in the particle surroundings as well as strong electromagnetic energy localization. In spite of the strong response of metallic materials, like gold and silver in infrared and visible spectral ranges, their inherent ohmic losses make them less attractive for some particular applications, among which those concerning optical communications[3].

High-refractive-index (HRI) dielectric particles with low absorption have been proposed as an interesting alternative to overcome these losses issues[4–11]. Some of their most important advantages are related to the fact that light can travel through them without being absorbed and their compatibility with well-known technologies as they can be made with either classical pure semi-conductors like silicon[5–7,9,11] and germanium[8] or semi-conductor compounds[4]. In addition, they can be designed to control the direction of the scattered radiation. Under some specific conditions, known as Kerker conditions[12–18], scattered radiation obtained from a single HRI spherical particle can be concentrated either in the back or forward scattering regions. For the latter, it is even possible to produce a null scattering effect in the exact backward direction. These effects, which in the literature are called magnetodielectric, are a consequence of the coherent effects between the excited electric and magnetic dipolar modes. In general, whispering gallery modes are responsible for resonances involved in these phenomena, which can be either electric or magnetic, even though the particle is non-magnetic at all[5]. Their coherent contribution produces interference effects leading to peculiar directionality phenomena which can result in various applications for optical communication purposes. As such, very recently, it has been shown that small dielectric HRI spheres can be proposed as new multifunctional elements for building optical devices[19–21]. This can be achieved by taking advantage of the coherent effects not only between dipolar contributions but also between these and modes of higher order, giving rise to anomalous scattering effects[22]. These open an extra way to control the directionality of the scattered light.

By pursuing this idea, here we introduce the possibility of using a HRI dimer as an elementary unit for building in-plane binary switching devices. Although this possibility has also been recently proposed[23] with a dark-field microscope configuration, back and forward directions are not really suitable for practical purposes for building operational optical circuits. In the forward direction, the scattered (wanted) signal is mixed with the incident beam which usually has a much higher intensity. In this case, it is necessary to add an extra element (light shielding plate) to avoid direct radiation which can mask the real forward scattered radiation. This has been recently proposed in ref. 24 for analysing Fano-like phenomena with metallic nanostructures (transmission dark-field geometry).

In the backward direction, the scattered field is not easy to isolate without using some sort of beam-splitter (dark field configuration in optical microscopy) with the corresponding complication in designing the scattering arrangement and as it can be shown, it is not possible to get perfect switching even if the two spheres do interact. Therefore, back and forward directions need additional elements which complicate the optical configuration for building operational optical circuits. Thus other directions, different from back and forward, seem more appealing for practical in-plane optical circuits design. In this work, we show that the scattered intensity at 90° from the incident direction can be null or maximum by playing with the polarization of a single frequency excitation and with a dimer whose components are close enough to interact in a controlled way. This interesting behaviour converts the dimer element in a two-output (beam splitter) switching unit whose binary state depends only on the polarization of the exciting radiation. Here, the analysis of this polarization switching effect is performed through the determination of the polarimetric parameter $P_L(90°)$, that is, the linear polarization degree of the scattered radiation at a direction of detection which is perpendicular to the incident direction. Moreover, in order to finely understand the underlying physics associated to this polarization switching effect, we analytically study the different contributors by means of a Green's function formalism and a dipolar approximation[25]. As this is an efficient tool to describe the electromagnetic interaction between particles, it enables to highlight the key parameters which are responsible for the expected behaviour. We also demonstrate that this effect is perfectly tunable by simply changing the particle sizes. Finally, it is important to remark that although the experiments contained in this research are made in the microwave range, the scalability of Maxwell's equations permits generalizing their results to other dimensions and spectral ranges, including those of the nanoscale and the visible region. This paves the way to design and build new optical elements to perform logical operations with light.

## Results

**Experimental configuration.** The interaction effect between the two components of a homogeneous dimer has been analysed for two identical spherical particles of radius $R_1 = R_2 = R$ and made of an HRI material ($Re(n) > 3$). The considered particle is smaller than the illuminating radiation wavelength $\lambda$ (the size parameter $q$ is defined as $2\pi R/\lambda$). The dimer is oriented along the $y$ axis. This structure is excited by a linearly polarized plane wave propagating along the $z$ axis and polarized either parallel to the dimer principal axis (longitudinal configuration) or perpendicular to it (transverse configuration), as it is shown in Fig. 1. The gap separation distance between the two external boundaries of the

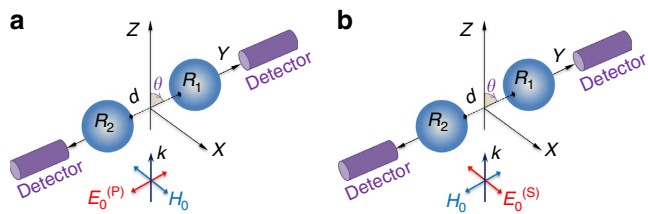

**Figure 1 | Scattering configuration.** A homogeneous sphere dimer of radius $R$ ($= R_1 = R_2$) is illuminated by a monochromatic plane wave propagating along the $z$ axis and linearly polarized either parallel to the dimer connecting axis: (**a**) longitudinal configuration (P), (**b**) transverse configuration (S). The gap distance is denoted by $d$ and the interaction parameter by $d_0 = d/R$. $\theta$ corresponds to the scattering angle.

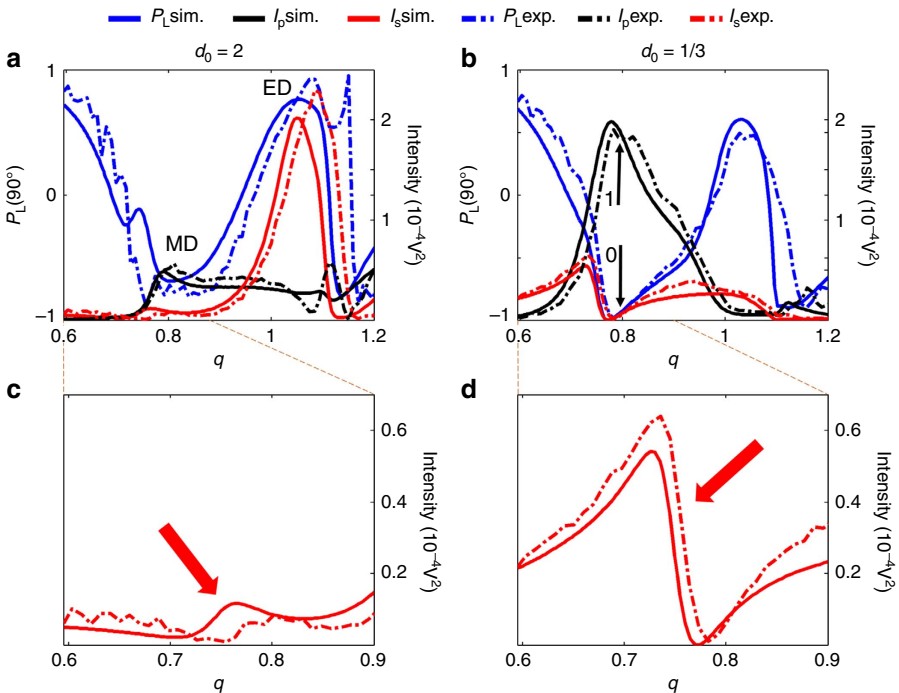

**Figure 2 | Polarimetric spectral measurements.** Linear polarization degree at right-angle scattering configuration $P_L$ (90°) and scattered intensities from simulations and experiments at 90° for a sphere dimer as a function of the size parameter $q$. The exciting radiation is linearly polarized and the electric field is either perpendicular ($I_S$) or parallel ($I_P$) to the scattering plane (see Fig. 1). (**a**) Simulation and measurement for $d_0 = 2$. (**b**) Simulation and measurement for $d_0 = 1/3$ (MD = Magnetic dipole resonance, ED = Electric dipole resonance). Simulated $P_L$ (solid blue line) and measured $P_L$ (dash-dotted blue line). Simulated $I_S$ (solid red line) and measured $I_S$ (dashed red line). Simulated $I_P$ (solid black line) and measured $I_P$ (dash-dotted black line). The low(off)/high(on) states are indicated with 0/1 and the black arrows. (**c**,**d**) are zoomed details for $I_S$ of plots (**a**,**b**) respectively to show how the natural magnetic resonance (marked with an arrow in **c**) evolves to a Fano-like shape (marked with an arrow in **d**) as the interaction between the components of the dimer increases.

spheres is denoted by $d$. In Fig. 1, the detection direction is fixed at $\pm 90°$ because, as we will see in the following, the performance of the switching effect is optimum for this configuration, that is, the difference between 'on' and 'off' situations is maximum. In order to express the gap distance in a dimensionless way, we define a new relative parameter, $d_0$, as the distance between the two particles, $d$, divided by the particle radius. This parameter is an indicator of the strength of the electromagnetic interaction between the two components of the dimer. Two gap distances are considered, $d_0 = 1/3$ (small gap→strong interaction) and 2 (large gap→weak interaction).

**Linear polarization degree for a high-refractive-index dimer.** The linear polarization degree, $P_L(\theta)$, can be defined[26–29] through the scattered intensities at a scattering angle $\theta$, when the exciting radiation is linearly polarized, oriented perpendicular and parallel to the scattering plane, as:

$$P_L(\theta) = \frac{I_s(\theta) - I_P(\theta)}{I_s(\theta) + I_P(\theta)} \qquad (1)$$

$I_S(\theta)$ ($I_P(\theta)$) is the scattered intensity at a scattering angle $\theta$, when the exciting radiation is linearly polarized and perpendicular (parallel) to the scattering plane ($ZY$ in Fig. 1), which corresponds here to the transverse configuration (longitudinal configuration). As it was demonstrated in previous studies, this polarimetric parameter $P_L$ is an efficient alternative to the more conventional extinction efficiency determination in light scattering experiments for particle sensing and sizing[13,30–32]. Furthermore, it can provide

information about either the electric or magnetic nature of resonances[13].

In Fig. 2, the spectral behaviour of $I_S(90°)$, $I_P(90°)$ and the linear polarization degree $P_L(90°)$ are plotted in the dipolar spectral region, where only the electric and magnetic dipole contributions exist (numerical (continuous)/experimental (dash-dotted)), and for the small- and large-gap cases respectively (see Methods). It is possible to distinguish the dipolar electric and magnetic resonances, labelled in Fig. 2a as ED and MD, respectively. Comparing the small- and large-gap cases, one of the most remarkable differences is the values of $P_L(90°)$ around $q = 0.8$. For the small-gap case, $P_L(90°)$ reaches $-1$, while for the large one it clearly deviates from this value. According to equation (1), it is clear that $P_L(90°) = -1$ corresponds to null values of $I_S(\theta)$ and, as shown in ref. 30 in the dipolar region, negative values of $P_L(90°)$ are linked to magnetic contributions. Around $q = 0.8$, as the distance between the particles decreases, the intensity $I_P$ increases while $I_S$ decreases. In particular, $I_S$ reaches null values for $d_0 = 1/3$ at $q = 0.773$ while $I_P$ reaches simultaneously its highest value. The reason of this behaviour is twofold as analysed by means of the Green's formalism and the dipolar approximation (Supplementary Note 1 and Supplementary Figs 1–11). First, the coupling between the two particles generates a supplementary induced magnetic dipole orientated along the propagation direction, when the transverse configuration is considered. Second, even if the induced electric and magnetic dipoles are less than $\lambda/2$ apart, they both generate electric fields which destructively interfere at $q = 0.773$ for 90° scattering angle. Indeed, at that frequency, the interference of the

electric field created by the electric and magnetic dipoles in both particles is destructive as their phase difference is $\pi$. Furthermore, the amplitude of the electric field created by the electric and magnetic induced dipoles in the first particle is the same as that created by the induced dipoles in the second particle. Thus, at that particular frequency, the dimer as a whole works as a pure magnetic unit ($P_L(90°) = -1$) leading to a perfect switching. It is also remarkable that this switching effect is produced by the spectral evolution of one of the natural resonances (the dipolar magnetic) of the isolated particle to an asymmetric shape resonance (Fano-like) as the particle interaction increases and the consequent implications in the practical use of this kind of spectral asymmetric shapes[33–36]. This means that the control of the electromagnetic interaction in a dimer element permits not only the proposal of a new binary (switching) unit but a control of the spectral properties of the scattered radiation. The origin of this Fano effect comes from the coherent interaction of the electromagnetic field due to the electric dipole resonance whose tail acts as the broad channel and that of the magnetic resonance acting as the narrow channel[37] (see Supplementary Note 1 and Supplementary Figs 6 and 7 for details)

The fact that, at the same spectral position, denoted in the following as the switching frequency, $I_P$ and $I_S$ take so different values has important consequences for switching purposes. When both components of the dimer are close enough to interact, the ensemble behaves as a polarization switching element. This switching behaviour is observed for different scattering angles. However, we operate at 90° for two main reasons. First, the maximum difference between the parallel and perpendicular to the scattering plane intensities is obtained for the right-angle scattering configuration, which means that a clear difference between two states of the switching device can be easily detected. Second, detecting at 90° is a good way to avoid any parasitic effect due to the incident radiation (considering a reasonably focused incident beam) making this dimer geometry and the incoming wave probably the best configuration for this switch to be of great practicality. Another remarkable feature is that we can obtain information about the charge distributions in the particles, observing the existence of a pure magnetic dipole at the switching frequency through the polarimetric parameter $P_L(90°)$. In general, we have an additional feature due to the symmetry of the electromagnetic problem: two identical outputs can be handled at the same time with the same scattering unit and for one incidence. This feature is similar to that of a beam splitter. Although this switching effect can be also observed for an isolated sphere, the dimer presents two main advantages. First, the difference between the high and low (or 'on' and 'off') states is higher than with a single sphere. Second, the low state is zero (without noise), which makes it easier to detect and the high state reaches higher values than for an isolated sphere.

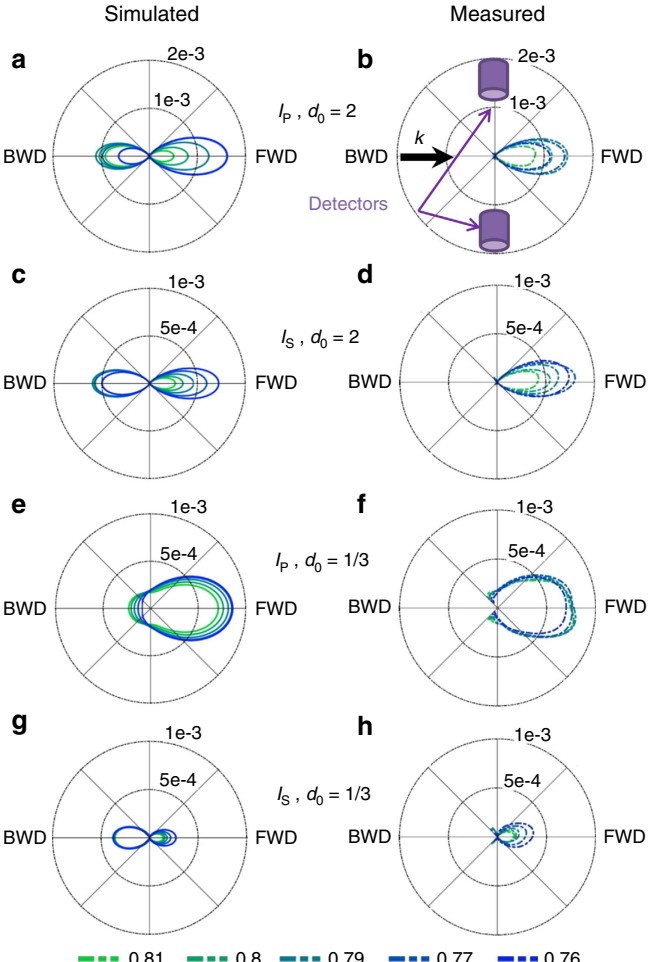

**Figure 3 | Angular variations of scattered intensities.** Polar plot of the scattering intensities in the scattering plane around the switching frequency $q = 0.773$: $q = 0.81$, 0.8, 0.79, 0.77 and 0.76 (see bottom line for colour code). (**a,b,e,f**): Longitudinal ($I_P$) configuration. (**c,d,g,h**): Transverse ($I_S$) configuration. (**a–d**): Large gap, $d_0 = 2$. (**e–h**): Small gap, $d_0 = 1/3$. (**a,c,e,g**): Simulated intensities. (**b,d,f,h**): Measured intensities. It can be seen that the measured angular range does not include backward scattering due to experimental restrictions (losing 100° backward). The black arrow indicates the direction of the incident radiation. The purple cylinders represent the detectors positions for the beam-splitter configuration. FWD stands for Forward Direction (0°) and BWD stands for Backward Direction(180°).

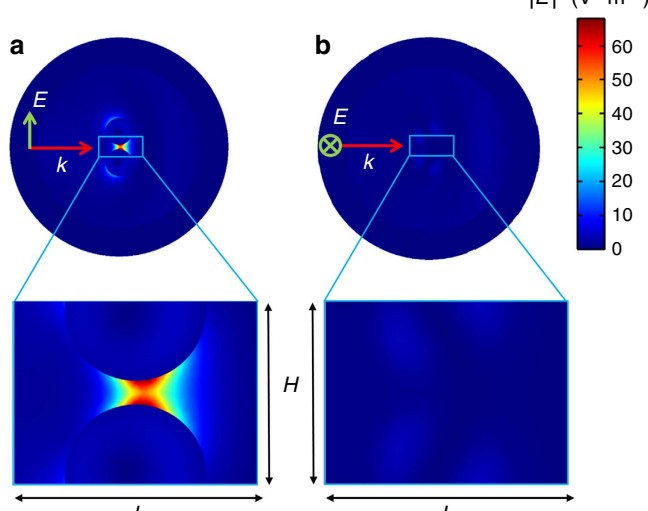

**Figure 4 | Near-field intensity distributions.** Near field numerical maps of the electric field intensity ($|E|^2 V^2 m^{-2}$ in linear scale) in the scattering plane ($ZY$ in Fig. 1) for the (**a**) longitudinal and (**b**) transverse configurations at the switching frequency $q = 0.773$. Only the small-gap case, $d_0 = 1/3$, is represented. The external annular domain corresponds to the perfectly matched layer. A closer view of the electric field intensity inside the gap is provided in the zoomed region corresponding to the blue rectangle drawn in **a,b**, ($L = 28$ mm, $H = 21$ mm).

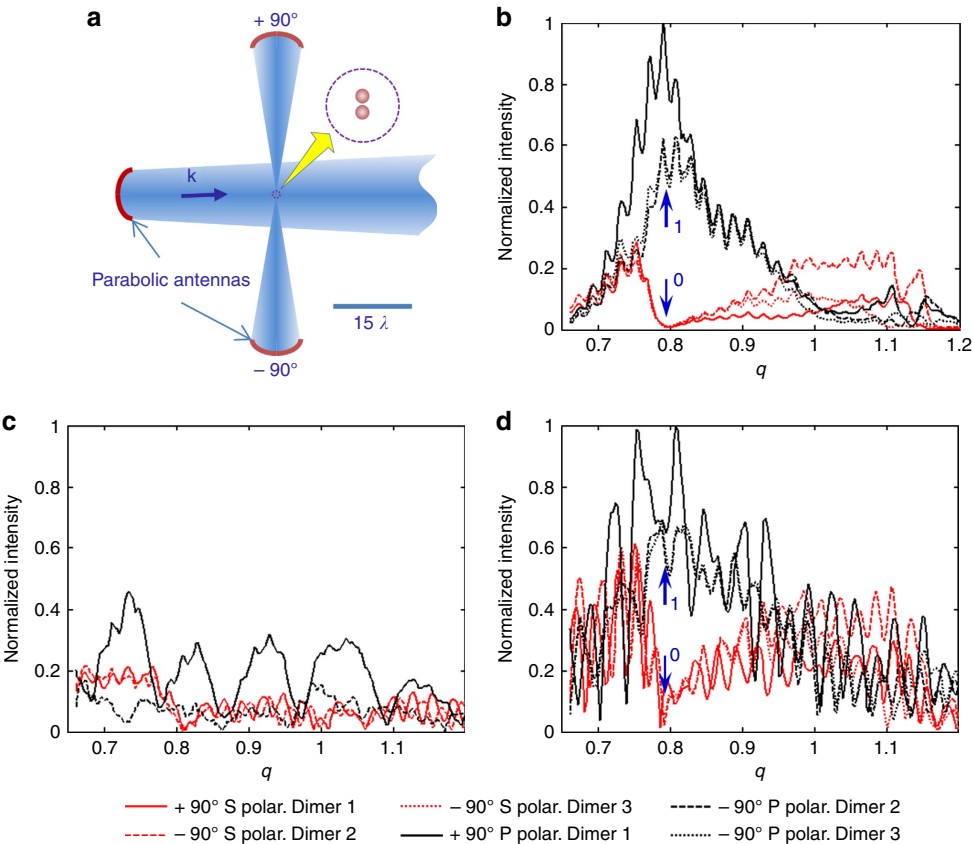

**Figure 5 | Direct measurements of the switch states.** (**a**) Experimental setup with directive antennas. (**b**) Raw measured scattered fields for receivers located at $\pm 90°$ and for three different dimers, Dimer 1, Dimer 2, Dimer 3 (corresponding to three different associations of four spheres: the two spheres previously used plus two other spheres made of the same material and with identical diameters). (**c**) Raw measured fields without any dimer to indicate the level of the residual signal due to the direct radiation between the antennas. (**d**) Raw measured field with a dimer ($d_0 = 1/3$). Direct detection of the S and P electric field at $\pm 90°$ without any (or at least very low) interaction with the incoming wave showing that the 'on' and 'off' states of the switching device can be directly measured without any reference nor processing.

In Fig. 3, the scattering diagrams for both polarizations and for the two gaps are shown in the scattering plane $ZY$ in the spectral range of the switching frequency. With the small gap, a large difference is observed between the scattered intensity values at 90° when comparing the two considered polarizations. However, for the large gap, the scattered intensities in the two polarization cases are both very small, which means that this switching effect is not observable at this wavelength when the particles do not interact.

To have a closer look at the field interaction in the vicinity of the dimer, the near field maps of the electric field intensity ($|E|^2$) are plotted in Fig. 4 for the small-gap case. A major difference in the intensity values is observed between the two polarization cases. For the longitudinal ($I_P$) polarization (Fig. 4a), a hot spot can be observed in the gap. However, for the transverse ($I_S$) polarization (Fig. 4b), the intensity values are negligible compared to the previous ones. The field evolution in near-field is thus in perfect agreement with the one observed in the far field and shown in Fig. 2.

In practical conditions, a switch must be built in order to be performing even within the most challenging conditions. With this in mind, the experiments shown in Fig. 5 have been made with a rather focused incoming wave (as a laser beam can be), in order to fully assess the potential of dimers as optical switches. Those experiments are presented here without any kind of processing and are definitely raw measurements, as they would be with an actual device. Indeed, Fig. 5c,d are showing the intensity acquired directly from the receiver when measured with or without the dimer (the signal is not even referenced to the signal delivered to the source antenna, and no frequency variation is compensated). Only Fig. 5b is obtained through the subtraction of these two raw complex measurements. Notice that the small differences between the measured fields in the three experiments with the P polarization observed in Fig. 5d are mainly due to the antenna's imperfection, which are visible in Fig. 5c. Figure 5d definitely provides a scientific evidence that a simple measurement of the intensity seen at $\pm 90°$ when changing the polarization of the incoming wave can undeniably be used as the 'on' and 'off' states of a switching device. Furthermore, even if those measurements are made in the gigahertz range, this last proof of concept made with raw intensities really paves the way to exploiting such a device in optics.

**Tuning the switching frequency.** One of the most important characteristics of a device is the reproducibility of its performance in different spectral ranges. In particular, for our switching device, the switching frequency is perfectly tunable to different wavelengths by only changing the size of the particles. In this section, we consider two different radii, $R_1 = R_2 = R = 6$ mm and $R_1 = R_2 = R = 12$ mm. The gaps $d$ are selected such that the gap-to-particle size ratios $d_0$ take the same values as the ones studied in the previous section. In Fig. 6 we represent $I_S$, $I_P$

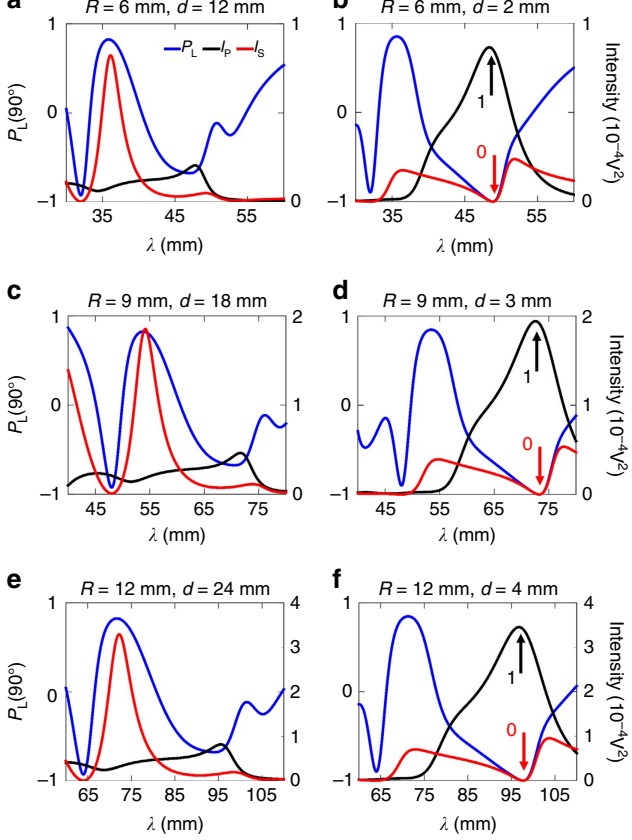

**Figure 6 | Spectral evolution of the switching frequency.** Linear polarization degree, $P_L(90°)$ (blue line and left vertical scale in each figure) and scattered intensities $I_S$ (red line and right vertical scale in each figure) and $I_P$ (black line and right vertical scale in each figure) measured at 90° in the scattering plane for a sphere dimer. Three types of dimer with identical spheres are considered: (**a,b**) $R_1 = R_2 = R = 6$ mm, (**c,d**) $R_1 = R_2 = R = 9$ mm and (**e,f**) $R_1 = R_2 = R = 12$ mm. Two gaps are analysed: (**a,c,e**) $d_0 = 2$ and (**b,d,f**) $d_0 = 1/3$. The black/red arrows represent the 'on' (1)/'off' (0) states of the switching device, respectively. Simulations are performed with the dipolar model (see Supplementary Notes 1 and 2). Inset in top-left plot indicates the colour code.

and $P_L(90°)$ for the new particle sizes, as well as for the previous one for the sake of comparison. Taking into account that the switching effect is observed in the dipolar region, the results shown are carried out using the analytical solution by means of the dipolar approximation without loss of generality. For the small gap, the particles are close enough to interact between them. For the small gap $d_0 = 1/3$, the switching effect is observed at shorter (longer) wavelengths for $R = 6$ mm ($R = 12$ mm) than for $R = 9$ mm. This behaviour is expected as the spectral position of the resonance depends on the particle size. As the particle size increases (decreases), the resonances are red (blue) shifted[38]. In fact, when the electric permittivity value is higher than 12, it can be shown that there is a single set of universal values ($d_0$, $q$) which will ensure the null value for the scattered intensity at 90° for the transverse configuration (see Supplementary Note 1). The radius value $R$ has thus a direct connection with the switching frequency range and can act as the tuning parameter to control in order to achieve the required switching frequency. Also, as $R$ increases, the switching high state level increases, leading to a more efficient logical device.

## Discussion

We have introduced theoretically and demonstrated experimentally a practical arrangement for using a homogeneous dimer of HRI dielectric spheres as a perfect binary polarization controlled switching device. This has been done by acquiring the scattered intensities at a right-angle scattering configuration. This can be considered equivalent to a 50/50 beam-splitter configuration which produces two identical beams from only one incident beam. It has also been shown that this arrangement leads to the highest offset between the two polarization switching states (in our case, 0 state corresponds to null intensity since the dimer behaves as a perfect magnetic scatterer), while being out of the more inconvenient but classical forward and backward directions which are fully impractical. We have also shown that its binary state depends only on the polarization of the exciting radiation. The possibility of tuning the switching frequency at which this phenomenon is the most significant has also been analysed. A wide spectral tuning range can be obtained by modifying the particle size of the dimer components.

Moreover, under the dipolar approximation, an analysis based on the linear polarization degree and on the Green's function formalism has enabled us to understand the physics of this electromagnetic problem and to optimize its behaviour. First, the presence of a supplementary magnetic dipole generated by the coupling effect between the two particles and second, the interference between the electric and magnetic dipoles induced by the incident beam in both particles of the dimer. We have also reported for the first time a pure magnetic dipole behaviour at the specific switching frequency, that is, at the right angle configuration the dimer as a whole works as a purely magnetic unit. This is the physical reason leading to a perfect switching response. The agreement between numerical calculations and experimental measurements is remarkable.

This binary switching configuration with electromagnetically interacting HRI dimers opens the way to generate new and practical optical devices. Although our experiments have been performed in the microwave range, the conclusions can be extended to the optical range due to the scalability of Maxwell's equations. Our research should stimulate experimentalists in the optical domain to overcome the technical difficulties of rescaling the dimer size and consequently the fabrication of these units. In this sense, at present, the fabrication of Si spheres has been technically improved and it is possible to get quite good spherical Si NPs with very low polydispersity[6,39]. However, the challenge of building good sphere dimmers with controlled separation still remains although there are some recent good examples in the literature, like those in refs 23,40. Other geometries like either cubes[41] or cylinders[11,42] could constitute suitable alternatives for building this kind of dimer units. Also, with reference to the experiment in the optical range, the widely used backward dark field microscope configuration (BDFMC) (see for instance, refs 17,43 among others) could be used to reproduce the results contained in this research, although our findings are mainly addressed to a final in-plane optical-board set-up. The lack of good directionality for both the illuminating and collected scattered radiation together with some difficulties in defining the polarization of the incident beam with respect to the dimer orientation represent technical problems of the BDFMC. These can weaken the switching effect (see Supplementary Note 2 and Supplementary Figs 9 and 12–15 for a detailed sensitivity analysis with respect to the parameters involved in this electromagnetic problem). As a first approach, they could be overcome respectively by means of masks to direct and collect light in specific directions (see for instance ref. 44) and the use of external arrangements (as that shown in ref. 45) in order to have a perfect control of the

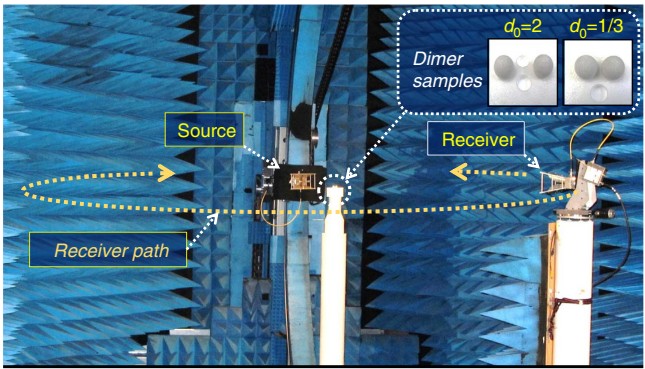

**Figure 7 | Experimental setup.** Picture of the experimental microwave setup in the anechoic chamber of the CCRM and the two HRI spheres in both large- and small-gap cases (in the inset). The source has a fixed position; the receiver moves circularly around the spheres which are positioned on the central vertical polystyrene mast. The scattered field measurement is performed in the horizontal scattering plane containing the source, the receiver and the target (spheres). The configuration in the photograph corresponds to the perpendicular polarization ($I_S$) case, here the transverse polarization configuration. The parallel polarization ($I_P$) case, here the longitudinal configuration, is obtained by rotating the source and the receiver.

exciting beam polarization. In summary, a combination of a modified microscope dark-field backward scattering configuration and an external oblique incidence for controlling the polarization of the incident light could be arranged in an optical experiment with BDFMC to reproduce a 90° scattering configuration. With our contribution, we want to leave doors open for new research in this kind of optical elements and the corresponding experimental arrangements. This includes the possibility of building arrays of the proposed unit for enhancing the effect.

## Methods

**Finite-element modelling.** From the numerical point of view, the results are obtained by means of a finite-element method implemented in the commercial software COMSOL Multiphysics[46]. In particular, we use the Radio Frequency Module that allows us to formulate and solve the differential form of Maxwell's equations (in the frequency domain) together with boundary conditions. We thus take advantage of the far-field pattern computation model (see Supplementary Note 1 for more details). The dimer is placed at the centre of a spherical homogeneous region filled with air, whose radius is $\lambda/2 + 2R$. A perfectly matched layer domain, with thickness $\lambda/4$, is positioned outside of the embedding medium domain and acts as an absorber for the scattered field. The mesh is chosen sufficiently fine as to allow numerical convergence of the results. In particular, the element size of the mesh of the embedding medium is smaller than $\lambda/5$ and that of the particles is smaller than $\lambda/[3Re(n)]$.

**Green's function model.** In order to understand the underlying physics more deeply, we also theoretically separate the different terms that contribute to the scattered intensity thanks to a Green's function formalism. In that case, each component of the dimer is modelled by an electric dipole and a magnetic dipole which are at right angles to each other as well as perpendicular to the propagation direction of the incident wave. More details on the Green's function formalism and the dipolar approximation for the transverse configuration can be found in the Supplementary Note 1, while the derivation for the longitudinal configuration is provided in ref. 25.

**Experimental methods.** Using the scale invariance rule, the so-called microwave analogy allows us to measure and characterize in the microwave domain phenomena which are very promising in the optical domain, such as the sought-after switching effect. Thus the experiments described below are totally similar to those that could be observed with dimers of silicon (Si) in the VIS-NIR range with dimensions of hundreds of nanometers. Our experiments were carried out using the microwave measurement facility in the anechoic chamber of the Centre Commun de Ressources en Microondes (CCRM) in Marseille, France. Over the last decade, this facility has interestingly become a specialized microwave

scattering device to perform analog to light measurements on a variety of complex particles[47]. The scattering measurements on a single HRI subwavelength sphere have recently been implemented to experimentally demonstrate the directional tunability of scattering radiations[19] and allowed the experimental proof of the Kerker conditions[13]. This experimental setup was used in this work to measure the scattered electric field, in both magnitude and phase and in parallel and perpendicular polarizations, by two analog-to-particle HRI spheres (Fig. 7).

The two HRI spheres are set at the centre of a horizontal disk of around 4 m in diameter. They are placed on a vertical polystyrene mast, almost transparent to electromagnetic waves and illuminated by an incident plane wave emitted from a fixed-position source. Both spheres are made of the same low-loss HRI material (HIK 500F from Laird Technologies) with $R_1 = R_2 = R = 9$ mm and $\varepsilon = 15.7 + 0.3i$. This permittivity was determined at the working frequencies (from 3 to 9 GHz) from far-field-scattering measurements acquired with only one of the spheres and through comparison to Mie calculations, following the technique described in ref. 48. First, the scattered field at each receiving position is used to determine one permittivity value over all the frequencies, as the permittivity has been observed to have insignificant variations with respect to the frequency. Afterwards, out of the receiving positions where the measurements are of lowest errors and lowest sensitivity to noise, one averaged permittivity is estimated and is used in all the numerical simulations. In order to control the gaps between the spheres (3 and 18 mm representing the small and large gaps respectively), a polystyrene sphere holder was especially fabricated for this purpose.

The experiments are not limited to the scattering measurement at 90° but are advantageously performed over an angular excursion of 260° around the target. This allows us to apply an angular-based post-processing procedure, important to reduce drift and noise effects and to enhance the data quality[49]. We draw the attention here to the challenges associated with this measurement since the spheres have small sizes compared to the wavelength (small $q$), the scattered intensity is relatively low thus very sensitive to random noise, drift problems and parasite signals. Notice that all the measurements presented here (apart from Fig. 5) are calibrated, which means that all the plotted intensities and values are truly quantitative ones (see Supplementary Methods for more details).

**Data availability.** All relevant data of this research are available from the authors.

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

## Acknowledgements

This research has been supported by MINECO (Secretaría de Estado de Investigación, Desarrollo e Innovación) through project FIS2013-45854-P and Fundación IBER-DROLA-ESPAÑA through its *Research on Energy and the Environment Program.* A.I.B. wants to thank the University of Cantabria for her FPU grant. We also acknowledge the opportunity provided by the Centre Commun de Ressources en Microonde to use its fully equipped anechoic chamber and for financing H.S. PhD grant.

## Author contributions

J.-M.G., A.I.B., H.S. and F.M. conceived this work. A.I.B., F.G., A.L. and F.M. developed the concept. J.-M.G., F.G., H.S. and F.M. conceived the experimental realization in the microwave regime. A.I.B., A.L. and F.M. performed the theoretical background. J.-M.G. and H.S. designed the experimental setup and performed the experiments. J.-M.G., A.L. and H.S. developed the post-processing treatments of the experimental data. H.S., A.I.B., J.-M.G., A.L., F.G. and F.M. analysed the experimental data. A.I.B., A.L., J.-M.G. carried out numerical calculations and figures. A.I.B., A.L., J.-M.G. and F.M. wrote the paper. All authors contributed to scientific discussion and critical revision of the article. F.M. and A.L. supervised the study. A.I.B. and H.S. have contributed equally to this research.

## Additional information

**Competing financial interests:** The authors declare no competing financial interests.

