## [Peer Review File · Nature Communications]

Reviewers' comments:

Reviewer #1 (Remarks to the Author):

The authors reported all-dielectric HRI dimer as new multifunctional elements for building optical switching devices. They used right angle scattering configuration rather than the widely used back and forward directions to realize optical switching. Although the physical mechanisms of magnetodielectric behavior in dimer have been investigated, the experimental characterization and the proposed "beam splitter" are interesting. Thus, I suggested reconsidering in Nature Communications after major revisions noted below.

(1) The title is a little bit confusing. Usually, only one sentence is allowed in title, so I recommend removing "THE BEAM-SPLITTER CONFIGURATION".

(2) The authors discussed the disadvantage of using forward/backward scattering as "switching" in Introduction. In fact, the forward and backward scattering light can be easily collected using dark-field objective. More reasons should be given on why using this kind of side scattering (90°).

(3) The authors said "the proposed switching device show more advantages than the devices based on the directional scattering between forward and backward", but they didn't demonstrate the scattering light between any two directions is different. They only showed that the scattered intensity at only one direction can be null or maximum by changing the polarization of a single frequency excitation, which has nothing to do with directional scattering.

(4) The authors reported the different optical responses under excitation with different polarization directions. These phenomena have already been reported in the plasmonic nanostructures (Nano Lett. 2012, 12, 4977). Thus, the differences should be clarified.

(5) The statement that "this switching effect is produced by the spectral evolution of one of the natural resonances of the isolated particle to an asymmetric shape resonance (Fano-like)" needs to be demonstrated. Where is the Fano line shape? Where are the broad mode and the narrow mode to produce Fano dip? All need to be marked in spectra.

(6) In Fig. 5, why the data are so fluctuant needs to be explained. Why measurements without any dimer (see Fig. 5c) also exhibit the "on" and "off" states?

(7) The "Supplementary Information" referred in the manuscript should have serial number.

Reviewer #2 (Remarks to the Author):

This paper presents a thorough and well-written description of a scheme to control the switching effect that occurs in the orthogonal direction to the incident propagation field illuminating a high refractive index dimer. This dimer composed of two dielectric spheres of equal radius is aligned to the direction in which the switching effect occurs, i. e., perpendicular to the incident propagation field. The switching effect is controlled by the polarization of the incident field. All the theoretical details are well supported by extensive and relevant literature, which includes several good papers published by the authors, and are also supported by a careful experimental demonstration in the microwave region, whose results, due to the electromagnetic invariance rule, can be used to validate the present scheme in the optical domain. The results reported in this paper are interesting, original and sound enough to justify their publication in this Journal.

Minor corrections follow below.

In the main text:

1. In Fig. 1, theta should be oriented taking as reference the positive axis z, not the negative one, as illustrated. This must be coherent with the orientation of theta shown in the polar plots of Fig. 3.
2. In page 8, lines 3 and 4, "are indicated" appears twice.
3. In page 12, lines 4 and 5, it is 4a and 4b, not 4ac and 4bd.
4. In Figs. 6a-6f, it is d, not d0.

5. In page 20, line 11, it is: ".....implemented the scattering measurements on a single...." Delete "in".
6. Page 27, reference 39 should start in line14.

In the "Supporting Information" text:

1. Page 17, lines 13 and 14, the statement "In the finite element method" is inaccurate, because the definition given in (67) is common practice in the antenna field to plot the so-called radiation patterns.

End of comments.

Reviewer #3 (Remarks to the Author):

The authors expand previously reported work on scattering by a pair of high refractive index (HRI) dielectric structures with sub-wavelength size to directions normal to the input wave propagation. To the best of my knowledge the work is original and properly addresses prior literature.

The paper is well written and the contribution clearly presented.

The work is well structured and the results supported by simulation, theoretical analysis and measurements on a fabricated prototype at microwave frequencies.

The application potential in building switching devices is highlighted and experimental results in microwave frequencies are presented, while the results are proposed to be general in application all the way to optical frequencies due to the scaling properties of Maxwell equations.

- 1) it is recommended to further comment on the challenges in implementing these structures at optical frequencies, a) in terms of fabrication and b) in terms of size and the effect of the presence of multiple pairs of HRI elements as it is unlikely that a single pair only can be fabricated in optical frequencies. how does this generalize into an array of hri elements?
- 2) some further detail on the measurement setup could be appreciated. how does the gain and directivity and polarization purity of the transmit and receive antennas affect the measurement. this is also related to the sensitivity of the system in terms of various parameters such as the angle of incidence of the incoming wave, and the system dimensions.
- 3) some quantitative results on the amount of scattered intensity relative to the input signal power should be included and how this can be controlled or optimized, as this has a direct effect on the application potential of the proposed structure as a beam splitter, ie what would be the loss associated with the device.
- 4) how does the input power affect the performance of the system? it is understood that this is a linear system but as the theoretical model includes only electric and magnetic dipole expressions, how would this be affected when higher order multipoles become significant.
- 5) although the presented results in figure 4 and 5 already hint on this, it would be interesting to further comment on the sensitivity of the system dimensions in order to observe the desired phenomenon.

Responses to Reviewers' comments:

Reviewer #1

The authors reported all-dielectric HRI dimer as new multifunctional elements for building optical switching devices. They used right angle scattering configuration rather than the widely used back and forward directions to realize optical switching. Although the physical mechanisms of magnetodielectric behavior in dimer have been investigated, the experimental characterization and the proposed "beam splitter" are interesting. Thus, I suggested reconsidering in Nature Communications after major revisions noted below.

(1) The title is a little bit confusing. Usually, only one sentence is allowed in title, so I recommend removing "THE BEAM-SPLITTER CONFIGURATION".

As the title should contain the key aspects of this research, we would like to ask for keeping the title as it is because this feature ("beam-splitter") is very important and makes our proposal different to other previous configurations based on either forward or backward conventional directions. Our configuration allows having two separated and controlled beams by using a single incident one. Also, Nature Communications allows for titles of maximum 15 words. Ours is 14. In any case, we are open to further suggestions.

(2) The authors discussed the disadvantage of using forward/backward scattering as "switching" in Introduction. In fact, the forward and backward scattering light can be easily collected using dark-field objective. More reasons should be given on why using this kind of side scattering (90°).

For configurations located on flat substrates for building optical devices, the 90° configuration is very attractive as compared to back/forward arrangements. For backward directions and taking into account our application purpose [see point 3] in response to referee's question #4], the necessity of introducing a beam-splitter (basically, this is the concept when dark-field microscopy is used) would complicate too much the device. For the forward direction, this is completely useless for practical purposes unless also, an additional element is added to avoid the incident beam which could mask the wanted signal. In summary, in these two specific directions, the use of additional elements (optical (beam-splitter) or not (mask or equivalent)) is necessary to play with the desired scattered radiation.

Also, it is important to mention that the wanted effect is optimum at 90° as it is shown in the figure R1 below. “0” corresponds to forward direction and “180” to backward direction. The blue line indicates the position of the switching frequency at $q \approx 0.77$. A sentence has been added in the main text to clarify this point.

Figure R1: Spectral evolution of the ratio (in dB) between I_s and I_p with respect to the measurement angle

(3) The authors said "the proposed switching device show more advantages than the devices based on the directional scattering between forward and backward", but they didn't demonstrate the scattering light between any two directions is different. They only showed that the scattered intensity at only one direction can be null or maximum by changing the polarization of a single frequency excitation, which has nothing to do with directional scattering.

It is well known that the light scattered in two different directions will behave differently (see for example Ref 19 for some measurement results for a single HRI sphere as well as Figure R1 above where the intensities are plotted with respect to the detection angle). The idea here is not to exploit two different detection directions, but instead to work with a single one and take advantage of the intensity variation in polarization. Among the various detection directions, one could use the forward or the backward one, but as well the 90° detection direction. In the supplementary material part, we explain the theoretical reason for favoring the 90° direction. Additional decisional elements are also provided in our answer to the previous point (2).

(4) The authors reported the different optical responses under excitation with different polarization directions. These phenomena have already been reported in the plasmonic nanostructures (Nano Lett. 2012, 12, 4977). Thus, the differences should be clarified.

1) We are using dielectrics with very low losses. In the reference indicated by the referee, the optical components of the proposed device are metallic with the inherent ohmic losses.

2) The proposed device in our research is simpler. Only two components are necessary with the corresponding implications in the device manufacturing. This difference exists since the technical objectives are completely different for the two configurations (see next point).

3) Its practical purpose is different to that in the above mentioned reference. Whilst in Nano Lett. 2012, 12, 4977, the goal is to build a "switching pixel" for which the control of the forward scattering is crucial, our device proposal has the objective of serving as an optical element built on a substrate (a board similar to that of an electric circuit) to redirect light in directions other than the unwanted (for our purpose) back and forward directions by keeping them parallel to the substrate (guiding light).

4) It can be shown that the switching effect is for a HRI dielectric dimer and at the 90° configuration it is "perfect" in the sense that a real "0" for the "off" position is possible.

Apart from these clarifications, we have made explicit reference to the suggested article in the introductory part of the manuscript. We believe that the corresponding comments have enriched the discussion about the differences (advantages/disadvantages) in either the backward or forward configurations with respect to the right-angle one proposed in our research. We want to thank this reviewer for this comment.

(5) The statement that "this switching effect is produced by the spectral evolution of one of the natural resonances of the isolated particle to an asymmetric shape resonance (Fano-like)" needs to be demonstrated. Where is the Fano line shape? Where are the broad mode and the narrow mode to produce Fano dip? All need to be marked in spectra.

We would like to thank this reviewer for these considerations. We have included the responses in the manuscript which, we believe, have improved considerably the interpretation of the Fano-like effect and consequently have enriched very much the Physics of our manuscript.

Fig. 2 has been replotted for showing the effect more clearly. Some comments have been added to the introduction to make clear the physical reasons of the appearance of the Fano-like effect. The corresponding full model and new figures have been included in the Supplementary Information. We do not detail them here for brevity.

(6) In Fig. 5, why the data are so fluctuant needs to be explained. Why measurements without any dimer (see Fig. 5c) also exhibit the "on" and "off" states?

To answer this question new measurements were made with attenuators to reduce the oscillations (due to resonances)..As the parabolic antennas are indeed designed to work in the 4-8 GHz range (i.e. $0.8 < q < 0.15$) an attenuator between the antenna and the receiver reduces some of the mismatching drawbacks and thus the oscillations (see Fig 5bis).

Figure 5bis: Additional measurements performed as in Figure 5, except that attenuators have been introduced in the setup.

It is right that there is a minimum around $q = 0.8$ (figure 5c, S polar) which was also found in the new experiments presented here (fig 5bis c) but it is probably due to the imperfections of

our anechoic chamber (another minimum is also visible at $q = 1$, where the dimer returns about 0.3). So the behavior of the measured field at $q = 0.8$ without dimer is just a coincidence.

To help to have a better understanding of the behavior of our experiments we have plotted (fig. 5ter) the same measurements than those presented in Figure 5 but with horn antennas (as in Figure 2). In Figure 2, the results were presented with post-processing, here there are shown without any post-processing of the measured data as in Figure 5.

Figure 5ter: Additional measurements performed as in Figure 5, except that horn antennas have been introduced in the setup (which corresponds to the measurement configuration used for Figure 2).

There is no minimum at $q = 0.8$ in those measurements but as the antennas are not directive enough the fields measured with and without the dimer are almost the same. In this case, to show the switching effect, we used a subtraction of the two measured fields to extract the scattered fields but, such a subtraction of the complex fields is not of any practical interest (especially in optics). Indeed the experiments (with the parabolic antennas) shown in Fig.5 were

made to prove that, if the signal is sent with a sufficient directivity (which should not be a real problem in optics), even without any special care (raw field measurements are presented), the switching effect related to the polarization states is of real practical interest with a single dimer and is indeed directly measurable

(7) The "Supplementary Information" referred in the manuscript should have serial number.

We are really sorry but we did not understand this comment.

Reviewer #2

This paper presents a thorough and well-written description of a scheme to control the switching effect that occurs in the orthogonal direction to the incident propagation field illuminating a high refractive index dimer. This dimer composed of two dielectric spheres of equal radius is aligned to the direction in which the switching effect occurs, i. e., perpendicular to the incident propagation field. The switching effect is controlled by the polarization of the incident field. All the theoretical details are well supported by extensive and relevant literature, which includes several good papers published by the authors, and are also supported by a careful experimental demonstration in the microwave region, whose results, due to the electromagnetic invariance rule, can be used to validate the present scheme in the optical domain. *The results reported in this paper are interesting, original and sound enough to justify their publication in this Journal.*

Minor corrections follow below.

In the main text:

1. In Fig. 1, theta should be oriented taking as reference the positive axis z , not the negative one, as illustrated. This must be coherent with the orientation of theta shown in the polar plots of Fig. 3.

Done

2. In page 8, lines 3 and 4, "are indicated" appears twice.

Done

3. In page 12, lines 4 and 5, it is 4a and 4b, not 4ac and 4bd.

Done

4. In Figs. 6a-6f, it is d , not d_0 .

Done

5. In page 20, line 11, it is: ".....implemented the scattering measurements on a single...."
Delete "in".

Done

6. Page 27, reference 39 should start in line14.

Done

In the "Supporting Information" text:

1. Page 17, lines 13 and 14, the statement "In the finite element method" is inaccurate, because the definition given in (67) is common practice in the antenna field to plot the so-called radiation patterns.

The sentence has been rephrased. Finally, we would like to thank this reviewer for his/her suggestions concerning text and figures in our manuscript. The corresponding corrections have permitted to improve its redaction.

Reviewer #3

The authors expand previously reported work on scattering by a pair of high refractive index (HRI) dielectric structures with sub-wavelength size to directions normal to the input wave propagation. *To the best of my knowledge the work is original and properly addresses prior literature.*

The paper is well written and the contribution clearly presented.

The work is well structured and the results supported by simulation, theoretical analysis and measurements on a fabricated prototype at microwave frequencies.

The application potential in building switching devices is highlighted and experimental results in microwave frequencies are presented, while the results are proposed to be general in application all the way to optical frequencies due to the scaling properties of Maxwell equations.

1) it is recommended to further comment on the challenges in implementing these structures at optical frequencies, a) in terms of fabrication and b) in terms of size and the effect of the presence of multiple pairs of HRI elements as it is unlikely that a single pair only can be fabricated in optical frequencies. How does this generalize into an array of HRI elements?

We thank this reviewer for this constructive comment. Although our experiments have been performed in the MW region, it is true that their conclusions can be extended to the optical range due to the scaling property of Maxwell equations. Our goal is to stimulate experimentalists in the optical domain to overcome the technical difficulties of rescaling the size and consequently the fabrication of these dimer units. In this sense, at present we can say that the fabrication of Si spheres has been technically improved and it is possible to get very good spherical Si NPs (from the geometrical point of view) with very low polydispersity (see new added references 38 and 39). However, the challenge of building good sphere dimers with controlled separation still remains even there are some recent examples in the literature, like that of ref 23 and that of the new added reference 40. Concerning this, it would be easier to build other geometries like either cubes or cylinders (see new added references 41, 42 and 43) but in these cases the “off = null intensity” condition is not satisfied although the switching effect would be there. Following referee’s suggestion, we have added some comments in the

conclusion section at the end of the manuscript with the corresponding references in order to leave doors open to research in this topic.

2) Some further detail on the measurement setup could be appreciated. How does the gain and directivity and polarization purity of the transmit and receive antennas affect the measurement. This is also related to the sensitivity of the system in terms of various parameters such as the angle of incidence of the incoming wave, and the system dimensions.

Part of the elements to answer this question can be found in the answer given to Rev. 1 point (6). With the parabolas, the gain is high (the antenna provider is just indicating a “3dB aperture” equal to 15° at the switching frequency) and the switching effect can directly be measured (fig 5, 5 bis). With the horn antennas the gain is rather low (about 12 dB at the switching frequency) and we needed a complex subtraction of the fields with and without the dimer to show the switching effect (Fig. 2, Fig. 5 ter). Concerning the polarization purity, both antennas cross polarization isolation is around 20 dB.

In summary a general answer is rather difficult to give but the less directive the source is, the further from the dimer the detector has to be (to avoid to be “blinded” by direct incoming wave). And last but not least, the further the receiver is from the dimer, the more sensitive it has to be, in order to allow separating the 0 and 1 states. Thus the best use of a dimer as a switching device is probably to be made with a rather directive source, which shouldn't be a real difficulty in optics. The sensitivity to the incidence angle and dimensions is numerically studied in the SI (and a 15λ scaling bar is represented on figures 5 to give an idea of the dimensions of the used experimental setup).

3) some quantitative results on the amount of scattered intensity relative to the input signal power should be included and how this can be controlled or optimized, as this has a direct effect on the application potential of the proposed structure as a beam splitter, ie what would be the loss associated with the device.

All the presented intensities are quantitative ones (apart from Figure 5). They all have been calibrated by comparing the field scattered by a single metallic sphere and illuminated with a plane wave of amplitude 1 at the center of the coordinate system (see Refs 47, 48) with the field actually measured with such a calibration target. The calibration aspect is evocated at the end of the Supplementary material part. Thus, in the figures (apart from Figure 5), all the intensity scales are meaningful and directly show the level of measured signal that one can expect at 23λ from the dimer (at the switching frequency).

4) How does the input power affect the performance of the system? It is understood that this is a linear system but as the theoretical model includes only electric and magnetic dipole expressions, how would this be affected when higher order multipoles become significant.

As the referee suggests we are only interested in linear phenomena because non-linearity effects would be out of the scope of our research. At present, we have not enough information to judge how the input power would affect system performance. It would depend on the optical properties of the dimer constituents and of course, the input power.

Concerning the influence multiple higher orders than dipolar, new research challenges appear which are out of the goal of our contribution. We have started to analyze this influence and as a first step, I would recommend our recent publication already cited in ref. 19.

5) Although the presented results in figure 4 and 5 already hint on this, it would be interesting to further comment on the sensitivity of the system dimensions in order to observe the desired phenomenon.

As suggested, a sensitivity analysis has been included in the supplementary materials. We have changed the various geometrical parameters, in particular the angle of incidence (by tilting the spheres) as well as the detection angle and the parameters associated to the dimer itself. This study gives good indication that the desired phenomenon is quite stable with respect to all these parameters.

Reviewers' comments:

Reviewer #1 (Remarks to the Author):

The authors have made a careful revision. However, the issue below is still needed to further clarify.

As for the comparison between 90o configuration and back/forward configuration, the dark-field oblique-incident system needs to be further discussed. In some works (like Nat. Comm. 4, 1527 (2013) and Nanoscale 8, 5996 (2016)), oblique incident was used and the collected scattering light was not pure backward scattering or forward scattering. In those cases, the scattering can be collected simply without the influence of incident light. This dark-field forward/backward scattering combined with oblique incident can also realize in-plane beam splitter. Therefore, the authors should provide further discussion about this issue.

Reviewer #3 (Remarks to the Author):

The authors have provided a detailed response to my comments, including further discussion related to the fabrication of the dimers in optical frequencies and a detailed analysis of sensitivity of the obtained results to various system parameters.

It would be interesting to include a brief comment on the expected performance based on an array of such dimers, however this certainly qualifies as a different topic which is not essential for the technical quality of this work.

The authors provide sufficient detail including supplemental material and experimental setup details, to facilitate the reproduction of their work.

I have no additional comments.

Reviewer #1

The authors have made a careful revision. However, the issue below is still needed to further clarify.

As for the comparison between 90° configuration and back/forward configuration, the dark-field oblique-incident system needs to be further discussed. In some works (like Nat. Comm. 4, 1527 (2013) and Nanoscale 8, 5996 (2016)), oblique incident was used and the collected scattering light was not pure backward scattering or forward scattering. In those cases, the scattering can be collected simply without the influence of incident light. This dark-field forward/backward scattering combined with oblique incident can also realize in-plane beam splitter. Therefore, the authors should provide further discussion about this issue.

We would like to thank again this reviewer for his/her constructive comment which has enriched the discussion section of our manuscript and also can stimulate new optical experiments taking advantage of the scaling property of Maxwell equations. We agree in the fact that in the configurations indicated in the suggested references (the first one corresponds to our ref 17 and only the backward configuration is used in the new suggested reference), *oblique incidence* is applied and the collected scattered light is not pure forward/backward scattering. Also, the collected light has no influence of the incident beam.

According to the reviewer suggestion, and in our opinion, the backward dark field microscope configuration (BDFMC) could be more suitable than that of the forward one (see for instance, refs [17] and [45] among many others) for implementing an arrangement close to our 90° scattering configuration (although our findings are mainly directed to a final in-plane optical board set-up). This is mainly because of the presence of the substrate which could be an “obstacle” for the forward configuration. In general, the dark field objective lens can produce a quite directional incident beam ($\approx 10^\circ$), as it can be seen in the Fig. 5 of both references. With the BDFMC, a first problem appears because the sample is illuminated from different directions at the same time and the scattered light is collected in a quite wide cone, of the order of 70° for the numerical apertures used in those experiments. The integration of the scattered light in this wide angular range could weaken the desired switching effect (see Supplementary Note 2). This sort of problems could be overcome by using “masks” to direct and collect light in specific directions (see for instance new ref [46]). A second problem comes out for defining the polarization of the incident beam with respect to the dimer orientation. This could be avoided by setting-up an “external” arrangement as that shown in new ref [47] concerning the way of illuminating the sample in order to have a perfect control of its polarization. In summary, a combination of a “modified” microscope dark-field backward scattering configuration and an “external” oblique incidence for controlling the polarization of the incident light could be arranged in an optical experiment with DFMC to reproduce a 90° scattering set-up (let’s remind that for this scattering angle, the switching effect is optimum as we showed in our previous revision).

As these comments would correspond to an extension to the optical range of our experimental research, we have added a new paragraph in the discussion section at the end of the manuscript. Also, a “subtle” change has been done in the abstract to be consistent with the added discussion. New references have also been included to illustrate this (see new refs [46] and [47])

Reviewer #3

The authors have provided a detailed response to my comments, including further discussion related to the fabrication of the dimers in optical frequencies and a detailed analysis of sensitivity of the obtained results to various system parameters.

It would be interesting to include a brief comment on the expected performance based on an array of such dimers, however this certainly qualifies as a different topic which is not essential for the technical quality of this work.

The authors provide sufficient detail including supplemental material and experimental setup details, to facilitate the reproduction of their work.

I have no additional comments.

We thank this reviewer for his/her comments. According to them, we interpret that the manuscript does not need new modifications. We are going to consider this suggestion for further research on this topic.